# Graph Neural Diffusion with Adaptive Skip Connection

## Abstract

Neural message passing on graphs can suffer from the oversmoothing problem, where repeated aggregation of neighborhood information causes node embeddings to become indistinguishable. This issue is not confined to discrete Graph Neural Networks (GNNs); it also arises in continuous-depth GNNs, such as Graph Neural Diffusion (GRAND), where a diffusion process governs feature evolution. Current solutions often involve adding auxiliary data-dependent source terms or employing nonlinear dynamics, rather than relying solely on pure diffusion. In this work, we propose a simple yet powerful linear alternative: Graph Neural Diffusion with Adaptive Skip Connection (GRAND-ASC). Our framework equips the standard GRAND model with a skip connection to the initial node features, which by itself is sufficient to prevent oversmoothing. Furthermore, to increase our model's adaptability, we introduce a learnable time-dependent parameter that dynamically balances the trade-off between integrating neighborhood information and preserving a node's initial features. We provide a theoretical foundation for GRAND-ASC, proving its analytical well-posedness and the numerical stability of its approximations. Furthermore, we formally demonstrate that our dynamics mitigate oversmoothing by ensuring the Dirichlet energy remains bounded away from zero. Through a comprehensive set of experiments, we demonstrate that our model achieves competitive state-of-the-art performance on node classification tasks, with particularly strong results on heterophilic benchmarks where preserving node-specific information is crucial. The source code is available at: https://tinyurl.com/3n8r6nxn.

## 1 Introduction

Neural message passing Gilmer et al. (2017) forms the foundation of modern graph representation learning, serving as the core mechanism for aggregating neighborhood information in graph-structured data. The earliest GNN architectures drew inspiration from spectral graph theory Kipf & Welling (2017); Defferrard et al. (2016), utilizing graph Fourier transforms to extract structural patterns in the spectral domain. This feature extraction can be performed either discretely, as seen in GCN Kipf & Welling (2017), GraphSAGE Hamilton et al. (2017), and GAT Veličković et al. (2018), or continuously through diffusion processes, such as continuous GNNs Xhonneux et al. (2020), GRAND Chamberlain et al. (2021a), sheaf diffusion process Bodnar et al. (2022), and recently graph neural Ricci flow Chen et al. (2025).

However, a significant challenge for both types of GNNs is oversmoothing, where repeated aggregation causes all node embeddings to become indistinguishable Oono & Suzuki (2020); Cai & Wang (2020). Importantly, this issue is not confined to deep discrete networks; it can also arise in shallow ones Wu et al. (2023c). Moreover, continuous models based on pure diffusion processes, such as GRAND, inevitably converge to a constant vector that is independent of the input Thorpe et al. (2022), resulting in an oversmoothing problem.

To address this problem, several modifications have introduced mechanisms to preserve node-specific information. One prominent approach is GRAND++ Thorpe et al. (2022), which incorporates source terms derived from labeled node features to GRAND. This method treats the features of labeled nodes as trustworthy anchors, continuously injecting them into the diffusion dynamics as corrective signals. The Allen-Cahn Message Passing framework Wang et al. (2023) takes a different

approach, modeling node interactions through a nonlinear reaction–diffusion equation. This non-linearity introduces a competing reaction force that counteracts pure diffusion, naturally preventing oversmoothing and enabling the formation of distinct clusters.

Employing nonlinearity has been claimed as the only solution to resolving the oversmoothing problem Wang et al. (2025) in continuous message passing. In this work, however, we show that a simple yet powerful linear alternative exists: equipping the GRAND model with a skip connection to the initial features. The inclusion of this skip connection is, by itself, sufficient to counteract the convergence to a constant vector of pure diffusion and resolve oversmoothing. Nonetheless, improving the performance of the continuous models is generally difficult as they must use a single set of weights throughout the entire integration time. To overcome this limitation, we employ an adaptive mechanism by introducing a learnable, time-dependent parameter that dynamically balances the trade-off between aggregating information from neighbors and preserving the node's initial features. In this way, skip connections effectively resolve oversmoothing, while adaptivity further enhances model performance. The mechanism is parameter-efficient, as it only requires learning a single scalar value per layer, resulting in a negligible increase in the total number of parameters in comparison with GRAND. Our contributions can be summarized as follows:

- **Graph Neural Diffusion with Adaptive Skip Connection (GRAND-ASC)**: We propose GRAND-ASC. This simple yet powerful continuous architecture mitigates oversmoothing through a skip connection to the initial features. Crucially, we integrate this with a time-adaptive mechanism that dynamically balances two competing objectives: smoothing information from neighbors and retaining the node's original feature. This framework provides an adaptive and efficient strategy that eliminates the need for data-dependent source terms Thorpe et al. (2022) or nonlinear dynamics Wang et al. (2023).

- **Theoretical Guarantees on Analytical Well-Posedness and Numerical Stability**: We will prove that GRAND-ASC is well-defined, i.e., it admits a unique analytical solution and the solution remains bounded over time. We also provide the stability analysis for Euler and the fourth-order Runge-Kutta (RK4) approximation of the proposed diffusion dynamics. This theoretical foundation guarantees that our model's predictions are robust and not subject to wild fluctuations due to numerical errors, a critical assurance for practical deployment.

  **Mitigating Oversmoothing**: Despite its linearity and simplicity, we prove that the proposed dynamic method mitigates oversmoothing by showing the Dirichlet energy is bounded away from zero.

- **Empirical Validation on Heterophilic Graphs**: We demonstrate the effectiveness of GRAND-ASC through extensive experiments on the node classification task. These results demonstrate that GRAND-ASC is a robust general-purpose model that excels particularly in heterophilic settings while maintaining acceptable performance on homophilic graphs.

## 1.1 Graph Notation

Let $G = (V, E, \mathbf{W})$ be an undirected, weighted graph with node set $V$ of cardinality $|V| = n$, edge set $E$, and symmetric adjacency matrix $\mathbf{W} \in \mathbb{R}^{n \times n}$ where $[\mathbf{W}]_{ij} = w_{ij}$ represents the non-negative edge weight between nodes $i$ and $j$. The weight $w_{ij} = 0$ if $(i, j) \notin E$. We enumerate the nodes as $V = \{1, 2, \ldots, n\}$, with the neighborhood of node $i$ denoted by $\mathcal{N}(i) = \{j \in V \mid (i, j) \in E\}$. Each node $i$ has an associated feature vector $\mathbf{x}_i \in \mathbb{R}^d$, and the collective feature matrix is $\mathbf{X} = [\mathbf{x}_1, \ldots, \mathbf{x}_n]^\top$. The degree matrix $\mathbf{D} \in \mathbb{R}^{n \times n}$ is a diagonal matrix where each entry $[\mathbf{D}]_{ii} := d_i = \sum_{j \in V} w_{ij}$ represents the weighted degree of node $i$, corresponding to the sum of edge weights incident to node $i$. The graph Laplacian matrix, defined as $\mathbf{L} = \mathbf{D} - \mathbf{W}$, is a fundamental object in spectral graph theory. Its eigenvalues, denoted $\mu_1 \leq \mu_2 \leq \cdots \leq \mu_n$, reveal key structural properties of the graph. The normalized Laplacian matrix, defined as $\mathcal{L} = \mathbf{D}^{-1/2} \mathbf{L} \mathbf{D}^{-1/2}$, provides a scaled alternative whose eigenvalue spectrum is constrained. The eigenvalues $\gamma_1 \leq \gamma_2 \leq \cdots \leq \gamma_n$ of $\mathcal{L}$ satisfy $0 = \gamma_1 \leq \cdots \leq \gamma_n \leq 2$ Chung (1997). The second smallest eigenvalue, $\gamma_2$, is commonly known as the algebraic connectivity. This value quantitatively reflects the overall connectivity of the graph, where a larger $\gamma_2$ indicates a more strongly connected structure.

## 1.2 Diffusion on Graphs

The gradient operator $\nabla : \mathbb{R}^n \to \mathbb{R}^{|E|}$ maps node features to edge features

$$(\nabla \mathbf{x})_{ij} = x_j - x_i \quad \forall (i, j) \in E,$$

measuring feature variation across edges. The divergence operator $\text{div} : \mathbb{R}^{|E|} \to \mathbb{R}^n$ defines as

$$(\text{div}(\mathbf{X}))_i = \sum_{j \in \mathcal{N}(i)} w_{ij} \, \mathbf{X}_{ij},$$

and maps edge features back to nodes. The diffusion dynamic is written in standard matrix form as

$$\frac{\partial \mathbf{X}(t)}{\partial t} = \text{div} \left[ \mathbf{G}(\mathbf{X}(t), t) \, \nabla \mathbf{X}(t) \right], \tag{1}$$

where $\mathbf{G}(\mathbf{X}(t), t) = \text{diag}\big(a(\mathbf{x}_i(t), \mathbf{x}_j(t))\big) \in \mathbb{R}^{|E| \times |E|}$ is diagonal. Each diagonal entry scales the entire row of $\nabla \mathbf{X}(t)$ corresponding to its edge. Here $a : \mathbb{R}^d \times \mathbb{R}^d \to \mathbb{R}^+$ is a learnable function that assigns a diffusion strength to each edge.

## 1.3 GRAND-ASC Message Passing

The following equation defines the governing diffusion dynamics of GRAND-ASC:

$$\frac{\partial \mathbf{X}(t)}{\partial t} = \lambda(t) \, \text{div} \left[ \mathbf{G}(\mathbf{X}(t), t) \, \nabla \mathbf{X}(t) \right] + (1 - \lambda(t))(\mathbf{X}(0) - \mathbf{X}(t)), \tag{2}$$

where $\lambda(t)$ is a skip connection strength, shared across all nodes and features, at layer $t$. This formulation enables each node to dynamically balance between integrating information from its neighbors (the diffusion term) and retaining its initial features (the memory term). If $\lambda(t) = 1$ for all layers $t$, the model simplifies to the standard GRAND diffusion process. This flexible design allows GRAND-ASC to smoothly transition between pure diffusion and feature preservation via the learnable function $\lambda(t)$.

Let $\mathbf{X}(t) = [\mathbf{x}_1(t) \; \mathbf{x}_2(t) \; \cdots \; \mathbf{x}_d(t)]$ be the feature matrix at layer $t$, where $\mathbf{x}_k(t) \in \mathbb{R}^n$ is the $k$-th column representing the feature vector for the $k$-th dimension across all nodes. For simplicity of notation, we will drop the subscript $k$ and focus on a single feature vector $\mathbf{x}(t) \in \mathbb{R}^n$ in the subsequent analysis, with the understanding that the same dynamics apply independently to each feature dimension. Let $x_i(t)$ denote the $i$-th element of the vector $\mathbf{x}(t)$, representing the feature value at node $i$. Then, the GRAND-ASC dynamics follow the differential equation

$$\frac{\partial x_i(t)}{\partial t} = \lambda(t) \sum_{j \in \mathcal{N}(i)} a(x_i, x_j)(x_j(t) - x_i(t)) + (1 - \lambda(t))(x_i(0) - x_i(t)) \tag{3}$$

To weight the influence between nodes, the diffusivity is modeled with an attention function $a(\cdot, \cdot)$. We employ a multi-head scaled dot-product attention mechanism Vaswani et al. (2017), which computes the attention coefficient for an edge $(i, j)$ as

$$a(x_i, x_j) = \frac{\exp\left(\frac{(\mathbf{W}_Q x_i)^\top (\mathbf{W}_K x_j)}{\sqrt{d_k}}\right)}{\sum_{k \in \mathcal{N}(i)} \exp\left(\frac{(\mathbf{W}_Q x_i)^\top (\mathbf{W}_K x_k)}{\sqrt{d_k}}\right)},$$

where $\mathbf{W}_Q, \mathbf{W}_K \in \mathbb{R}^{d \times d_k}$ are learned projection matrices, and $d_k$ is the feature dimension per head. To enhance stability and representational capacity, we use $h$ independent attention heads, averaging their outputs to form the final attention weights.

Finally, GRAND-ASC architectures consist of three components: an encoder $\phi$, a differential equation solver, and a decoder $\psi$. The encoder maps the input features to the initial state via $\mathbf{X}(0) = \phi(\mathbf{X}_{\text{in}})$, while the decoder produces the final node embeddings as $\mathbf{Y} = \psi(\mathbf{X}(T))$. Using the matrix form of Equation (3), the differential equation solver is given as

$$\mathbf{X}(T) = \mathbf{X}(0) + \int_0^T \left[ \lambda(t)\big(\mathbf{A}(\mathbf{X}(t)) - \mathbf{I}\big)\mathbf{X}(t) + (1 - \lambda(t))(\mathbf{X}(0) - \mathbf{X}(t)) \right] dt, \tag{4}$$

where $[\mathbf{A}(\mathbf{X}(t))]_{ij} = a(x_i, x_j)$ is the attention matrix.

### 1.4 RELATED WORKS

**Message Passing in GNNs.** The feature extraction in GNNs can be performed either discretely, as seen in GCN Kipf & Welling (2017), GraphSAGE Hamilton et al. (2017), and GAT Veličković et al. (2018), or continuously through diffusion processes, such as continuous GNNs Xhonneux et al. (2020); Hariri et al. (2025); Eliasof et al. (2021); Finder et al. (2025), fractional differential equations Maskey et al. (2023), GRAND-based models Chamberlain et al. (2021a); Thorpe et al. (2022); Wang et al. (2023); Li et al. (2024b), sheaf diffusion process Bodnar et al. (2022); Hevapathige et al. (2025), and recently neural Ricci flow Chen et al. (2025). Continuous message passing is inspired by the framework of neural differential equations Chen et al. (2018), which has led to many follow-up works in the GNN field Avelar et al. (2019); Poli et al. (2019); Wu et al. (2023a); Rusch et al. (2022); Gallicchio & Micheli (2020); Lin et al. (2024); Yue et al. (2025).

**Oversmoothing.** A key challenge for GNNs is their depth limitations, as increasing layers often causes a performance drop in models like GCN Oono & Suzuki (2020) and GAT Wang et al. (2019); Wu et al. (2023b); Dong et al. (2021). This decline occurs because repeated neighborhood averaging makes node embeddings increasingly similar and eventually indistinguishable from one another. The problem was first identified by Li et al. (2018), who showed repeated Laplacian smoothing causes embeddings in a connected graph to converge. Subsequent works Oono & Suzuki (2020); Cai & Wang (2020) confirmed the energy function of embedding approaches zero with depth. Oversmoothing also affects continuous models, such as the early GRAND framework Thorpe et al. (2022). Subsequent approaches have tackled this issue in various ways: GRAND++ Thorpe et al. (2022) uses auxiliary source terms, and ACMP Wang et al. (2023) introduces nonlinear reactions. A closely related work is Li et al. (2024b), which also combines a fidelity term with a diffusion process from Fick's law. However, key differences distinguish our approach. First, we introduce a time-dependent function to balance these terms dynamically, unlike their fixed coefficients. Second, our attention mechanism utilizes initial features only, resulting in a linear differential equation that provides theoretical guarantees, such as numerical stability even for high-order differential equation solvers and a lower bound on the Dirichlet energy. Finally, our model is simpler, as it omits the second-order regularization term they use for 2-hop neighbors.

**Skip Connection.** Motivated by the success of skip connections in deep learning He et al. (2016), there is growing interest in their use for GNNs. Early work by Kipf & Welling (2017); Li et al. (2019) demonstrated that skip connections yield significant experimental improvements. Later, Liu et al. (2021) introduced message passing with adaptive embedding aggregation and skip connections, while Yang et al. (2022); Chen et al. (2023) proposed difference skip connections to help GNNs focus on residual information between initial and output features.

The use of initial skip connections was popularized by PPNP Gasteiger et al. (2019), which incorporated them into a GCN framework. This idea was later extended by GCNII Chen et al. (2020), which combined initial skip connections with identity mapping to enable deeper architectures. Recent work by Scholkemper et al. (2025) shows initial skip connections in PPNP mitigate oversmoothing, and Zhang et al. (2023) studies adaptive initial skip connections with layer-wise learnable skip strengths.

Similar methods that incorporate skip connections not only to the initial node features but also to combinations of intermediate layer embeddings have demonstrated strong performance, as seen in Jumping Knowledge Networks (JKNets) Xu et al. (2018), DeepGCN Li et al. (2019), Higher-Order Graph Convolutional Architectures (Mixhop) Abu-El-Haija et al. (2019), Deep Adaptive Graph Neural Networks (DAGNNs) Liu et al. (2020), and at R–SoftGraphAI Li et al. (2024a).

## 2 THEORETICAL FOUNDATION

We establish the theoretical analysis of the GRAND-ASC framework in two steps. First, in Subsection 2.1 we show that the dynamics Equation (3) are well posed and satisfy a min–max principle, which guarantees bounded solutions over time. Then, in Subsection 2.2 we analyze two numerical solvers, explicit Euler and the RK4, and prove that both are stable under practical step-size constraints. Lastly, Subsection 2.3 is devoted to showing that the Dirichlet energy of the GRAND-ASC is strictly positive for any depth.

## 2.1 WELL-POSEDNESS ANALYSIS OF GRAND-ASC

The existence and uniqueness of solutions to the GRAND-ASC dynamic follows from Picard's existence and uniqueness theorem for ordinary differential equations (see Perko (2013) for more details). The right-hand side of Equation (3) is Lipschitz continuous in $x_i$. This Lipschitz condition holds because: (1) the attention mechanism $a(x_i, x_j)$ is typically smooth (often softmax-based), and (2) the residual term $(1 - \lambda(t))(x_i(0) - x_i(t))$ is linear in $x_i$. Therefore, by Picard's theorem, there exists a unique solution $x_i(t)$ over the interval $[0, \delta_T]$ for some $\delta_T > 0$. The Min-Max principle established in Theorem 1 further guarantees that solutions remain bounded for all $t \geq 0$, allowing the unique solution to be extended indefinitely. The proofs of all theorems are provided in detail in Appendix A.

**Theorem 1.** *(Min–Max Principle) The solution to the GRAND-ASC satisfies the following bounds for all $i \in V$ and $t \geq 0$*

$$\arg\min_j x_j(0) \leq x_i(t) \leq \arg\max_j x_j(0).$$

## 2.2 NUMERICAL APPROXIMATION OF GRAND-ASC

Since obtaining an analytic solution for GRAND-ASC is challenging due to its time-dependent dynamics, we next turn our attention to numerical solvers. The goal here is to ensure that practical discretizations not only approximate the dynamics accurately but also preserve stability. We analyze two schemes. First, the explicit Euler method in 2.2.1, the most straightforward time approximation, whose special case recovers the classical GAT with an adaptive initial skip connection. We then analyze the RK4 method in 2.2.2, a higher-order solver with improved accuracy and a larger stability region. We show that both approximations remain stable for time steps $\Delta t \in (0, 1]$.

### 2.2.1 EXPLICIT EULER: REVISITING THE INITIAL SKIP CONNECTION IN GAT

Applying the explicit Euler method with step size $\Delta t$ to Equation (3) gives the following update rule

$$x_i(t+1) = x_i(t) + \Delta t \left( \lambda(t) \sum_{j \in \mathcal{N}(i)} a(x_i, x_j)(x_j(t) - x_i(t)) + (1 - \lambda(t))(x_i(0) - x_i(t)) \right). \quad (5)$$

The following proposition proves the numerical stability of this solver. This result guarantees control over the norm of the discrete solution and ensures that the method preserves the well-posedness of the continuous model.

**Theorem 2.** *Assuming for some $\delta > 0$, $\lambda(t) \leq 1 - \delta$ for any $t \geq 0$, the approximation (5) of GRAND-ASC is asymptotically stable for $\Delta t \in (0, 1]$, i.e., the sequence $\{\|\mathbf{x}(t)\|\}_{t \geq 0}$ is bounded by*

$$\limsup_{t \to \infty} \|\mathbf{x}(t)\| \leq \frac{1}{\delta} \|\mathbf{x}(0)\| \quad (6)$$

The assumption $\lambda(t) \leq 1 - \delta$, for some $\delta > 0$, is not practically limiting. In practice, these parameters are learned without restrictions, and we can apply a sigmoid transformation to ensure their values lie strictly between zero and one. This transformation naturally prevents the values from approaching too closely to either 0 or 1, making the assumption $\lambda(t) \leq 1 - \delta$ both realistic and achievable.

Setting $\Delta t = 1$ in Equation (5) and noting that the attention weights are normalized such that $\sum_{j \in \mathcal{N}(i)} a(x_i, x_j) = 1$, the diffusion term simplifies as

$$\sum_{j \in \mathcal{N}(i)} a(x_i, x_j)(x_j(t) - x_i(t)) = \sum_{j \in \mathcal{N}(i)} a_{ij} x_j(t) - x_i(t).$$

Thus, the explicit Euler scheme Equation (5), further simplifies to

$$x_i(t + 1) = \lambda(t) \sum_{j \in \mathcal{N}(i)} a(x_i, x_j) x_j(t) + (1 - \lambda(t)) x_i(0). \tag{7}$$

This recovers the GAT architecture with an initial skip connection (and if we set $\lambda(t) = 1$, then GAT Veličković et al. (2018) will be recovered), effectively forming a variant of initial skip connection (APPNP) Gasteiger et al. (2019); Scholkemper et al. (2025), where a GAT-based propagation mechanism replaces the original GCN and also $\lambda(t) = \lambda$ for all $t \geq 0$.

*Remark* 1. The time complexity of explicit Euler Equation (5) is $\mathcal{O}(T \cdot |E| \cdot d + T \cdot n \cdot d^2)$, where $T$ is the number of time steps, $|E|$ is the number of edges, $n$ is the number of nodes, and $d$ is the feature dimension. The $\mathcal{O}(|E| \cdot d)$ term arises from the edge-wise attention score computations and feature aggregations, while the $\mathcal{O}(n \cdot d^2)$ term comes from the linear transformations applied at each node. Additional skip connections only contribute $\mathcal{O}(n \cdot d)$, which is dominated by the $\mathcal{O}(n \cdot d^2)$ term.

### 2.2.2 FOURTH-ORDER RUNGE KUTTA APPROXIMATION

We now turn to a more accurate solver. In Theorem 4, we demonstrate that RK4 maintains stability while offering significantly higher accuracy than Euler. To this end, we begin by rewriting the continuous-time GRAND-ASC dynamics as

$$\frac{d\mathbf{x}(t)}{dt} = \mathbf{B}\mathbf{x}(t) + \mathbf{c}(t), \quad \text{where} \quad \mathbf{B} = \lambda(t)\mathbf{A} - \mathbf{I}, \quad \mathbf{c}(t) := (1 - \lambda(t))\mathbf{x}(0). \tag{8}$$

The corresponding RK4 discretization with step size $\Delta t$ is then given by (see e.g., Butcher (2016) for details)

$$\mathbf{k}_1 = \mathbf{B}\mathbf{x}(t) + \mathbf{c}(t), \quad \mathbf{k}_2 = \mathbf{B}\left(\mathbf{x}(t) + \tfrac{\Delta t}{2}\mathbf{k}_1\right) + \mathbf{c}(t), \quad \mathbf{k}_3 = \mathbf{B}\left(\mathbf{x}(t) + \tfrac{\Delta t}{2}\mathbf{k}_2\right) + \mathbf{c}(t), \tag{9}$$

$$\mathbf{k}_4 = \mathbf{B}\left(\mathbf{x}(t) + \Delta t\, \mathbf{k}_3\right) + \mathbf{c}(t), \quad \mathbf{x}(t+1) = \mathbf{x}(t) + \frac{\Delta t}{6}\left(\mathbf{k}_1 + 2\mathbf{k}_2 + 2\mathbf{k}_3 + \mathbf{k}_4\right).$$

Before proving the main theorem, we introduce a key theorem that will serve as the foundation for our stability analysis. This result establishes the exponential decay of matrix powers for matrices with spectral radius strictly less than one. This property is essential for bounding the cumulative error in the RK4 iteration. The proof, which appears in Appendix A, relies on Gelfand's formula to construct the decay rate $\gamma$ and the accompanying constant $C$.

**Theorem 3.** *Let $\mathbf{M}$ be a matrix with $\rho(\mathbf{M}) < 1$. Then there exist constants $C > 0$ and $0 < \gamma < 1$ such that for any $j \in \mathbb{N}$, $\|\mathbf{M}^j\| \leq C\gamma^j$.*

The stability of RK4 approximation follows in three steps: first, the discrete system Equation (9) is expressed via the RK4 stability function $R(z) = \sum_{k=0}^{4} z^k/k!$ applied to the matrix $\Delta t\mathbf{B}$. Second, we show that for any eigenvalue $\mu$ of $\mathbf{B}$, the scaled value $\nu = \Delta t\mu$ lies within the disk $\mathcal{D} = \{\nu \in \mathbb{C} : |\nu + 1| < 1\}$, which is contained in the RK4 stability region (verified at key points). This ensures the spectral radius $\rho(R(\Delta t\mathbf{B})) < 1$. Finally, Theorem 3 guarantees the exponential decay of the matrix power, leading to a bounded solution. Thus, we will have the following theorem.

**Theorem 4.** *The RK4 discretization of GRAND-ASC dynamics is asymptotically stable for any step size $\Delta t \in (0, 1]$.*

Hence, both the explicit Euler and RK4 methods are stable for $\Delta t \in (0, 1]$. Although RK4 requires four intermediate function evaluations per step compared to the single evaluation of Euler's method, it achieves a superior fourth-order accuracy of $\mathcal{O}(\Delta t^4)$, as opposed to Euler's first-order accuracy of $\mathcal{O}(\Delta t)$. This enhanced efficiency, combined with RK4's significantly larger stability region (as

shown in Subsection 3.1), makes it a superior choice. Although higher-order methods like fifth-order RK (RK5) or DOPRI5 are also applicable, their stability analysis and implementation follow the same principles as RK4. *However, our focus as solver is on the RK4 scheme, as this solver is often regarded as the optimal trade-off between speed and accuracy for multi-step solvers, see e.g., Butcher (2016).*

## 2.3 OVERSMOOTHING MITIGATION

In this section, we prove that the dynamics described in Equation (3), which form the core solver for the architecture in Equation (4), mitigate oversmoothing. This result is formalized in Theorem 6. Our analysis employs the Dirichlet energy as a measure of feature smoothness. For a node feature vector $\mathbf{x}(t)$, this energy is defined as $\mathcal{E}(\mathbf{x}(t)) = \mathbf{x}(t)^\top \mathcal{L}\mathbf{x}(t)$. A key property of this energy is given in the following theorem, which provides a lower bound based on the spectral gap $\gamma_2$.

**Theorem 5.** *Let $\mathcal{L}$ be the symmetric normalized Laplacian of a connected graph, with eigenvalues $0 = \gamma_1 < \gamma_2 \leq \cdots \leq \gamma_n$. Let $\mathbf{x}(t) \in \mathbb{R}^n$ such that $\mathbf{x}(t)^\top \mathbf{D}^{1/2}\mathbf{1} = 0$ (i.e., $\mathbf{x}(t)$ is centered). Then*

$$\mathcal{E}(\mathbf{x}(t)) \geq \gamma_2 \|\mathbf{x}(t)\|^2.$$

This result follows from spectral graph theory by expanding $\mathbf{x}(t)$ in the orthonormal eigenbasis of $\mathcal{L}$. The centering condition ensures orthogonality to the first eigenvector $\mathbf{v}_1 = \mathbf{D}^{1/2}\mathbf{1}$ (corresponding to $\gamma_1 = 0$), forcing the expansion to use only eigenvectors with eigenvalues greater than $\gamma_2$.

The following theorem shows that the energy function of the GRAND-ASC is lower-bounded by a strictly positive value.

**Theorem 6.** *Assuming for any $t \geq 0$, $\langle \boldsymbol{x}(0), \boldsymbol{x}(t) \rangle > m$ for some $m > 0$ and, $1 - \lambda(t) \geq \delta$ for some $\delta > 0$, then for any mean-centered vector $\boldsymbol{x}(t)$, the Dirichlet energy of GRAND-ASC Equation (3) satisfies*

$$\mathcal{E}(\mathbf{x}(t)) \geq \frac{\delta m \gamma_2}{2d_{\max} + 1},$$

*where $d_{\max}$ is its maximum degree of the graph.*

The proof is established by transforming the node-wise system Equation (3) into an expression involving graph edges and the memory term. The diffusion component is then bounded by relating the sum over edges to the quadratic form of the graph Laplacian and applying the Gershgorin Circle Theorem to connect its influence to the maximum degree $d_{\max}$. Concurrently, the memory term is controlled via the inner product $\langle \mathbf{x}(0), \mathbf{x}(t) \rangle$. These bounds are synthesized into a key differential inequality for the squared $\ell_2$-norm, which is solved explicitly using an integrating factor technique to derive a time-dependent lower bound for $\|\mathbf{x}(t)\|$. Finally, asymptotic analysis of this solution, combined with Theorem 5, yields the desired uniform lower bound on the Dirichlet energy. *It should be noted that requiring a positive inner product $\langle \mathbf{x}(0), \mathbf{x}(t) \rangle$ in Theorem 6 ensures the angle between initial and propagated embeddings lies in $(-\pi/2, \pi/2)$. This aligns with homophily by maintaining node similarity over time, whereas a negative value would imply divergence.*

In the special case where $\lambda(t) = \lambda$ is constant for all $t$, we obtain the following result.

**Corollary 1.** *For GRAND-ASC dynamics with constant $\lambda(t) = \lambda \in (0, 1)$, then*

$$\mathbf{x}(t) \to (1 - \lambda)\big(\lambda\mathcal{L} + (1 - \lambda)\mathbf{I}\big)^{-1}\mathbf{x}(0) \quad as \quad t \to \infty.$$

This result indicates that the solution does not converge to a constant vector but instead depends on both the graph structure (through the Laplacian $\mathcal{L}$), the parameter $\lambda$, and also the initial feature. Consequently, the node features will not become identical across all graph nodes under the GRAND-ASC dynamics.

## 3 EXPERIMENTS

In Subsection 3.1, we employ a synthetic graph structure to demonstrate that GRAND-ASC mitigates oversmoothing (and to illustrate the larger stability region of the RK4 numerical solver). Then, in Subsection 3.2, we focus on node classification tasks and compare GRAND-ASC against several well-known discrete and continuous GNNs.

### 3.1 SYNTHETIC SETUP

We analyze the Dirichlet energy on a synthetic undirected stochastic block model with $100$ nodes divided into two classes. Node features are two-dimensional, drawn from normal distributions with means $\mu_1 = -0.5$, $\mu_2 = 0.5$, and standard deviation $\sigma = 1$. The connection probabilities are $p = 0.9$ (within-class) and $q = 0.1$ (between-class), a setup also employed in Wang et al. (2023). As shown in Figure 1, standard GNNs (GCN, GAT, SAGE) and GRAND suffer from oversmoothing, with their Dirichlet energy decaying to zero quickly. In contrast, GRAND-ASC's energy stabilizes at a positive level, validating Theorem 6. This figure also illustrates the stability regions for explicit Euler, RK4, and RK5 solvers for $\Delta t = 0.5$, a behavior representative of other datasets. A larger stability region (for the same time step constraints) for RK schemes is a reason for their superiority over the explicit Euler method.

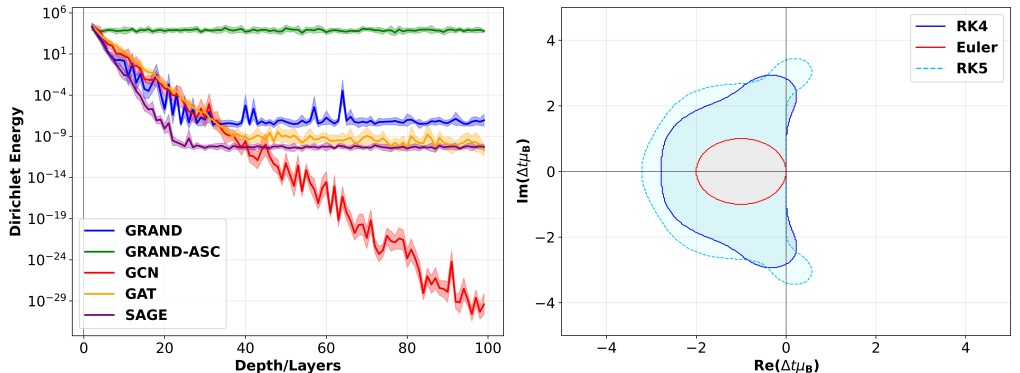

Figure 1: (Left) Mean Dirichlet (log scale) across layers for various models; the shaded region denotes the standard deviation. (Right) Stability regions for the Explicit Euler, RK4, and RK5 solvers, plotted in the complex plane where $\mu_{\mathbf{B}}$ is an eigenvalue of $\mathbf{B}$.

### 3.2 NODE CLASSIFICATION

#### 3.2.1 OTHER METHODS AND SETUP

We conduct a comprehensive evaluation of GRAND-ASC, comparing it against a range of baseline methods across multiple categories. This includes classical discrete GNNs such as GCN Kipf & Welling (2017), GAT Veličković et al. (2018), GraphSAGE Hamilton et al. (2017); skip connection-based architectures like Mixhop Abu-El-Haija et al. (2019), JKNet Xu et al. (2018), GCNII Chen et al. (2020), and APPNP Gasteiger et al. (2019); other state-of-the-art models including GraphGPS Rampášek et al. (2022) and the heterophily-focused DIRGNN Rossi et al. (2024); and finally, continuous message passing models such as CGCN Xhonneux et al. (2020), GRAND Chamberlain et al. (2021a), GRAND++ Thorpe et al. (2022), ACMP Wang et al. (2023), BLEND Chamberlain et al. (2021b), and NSD Bodnar et al. (2022).

We assess our models on the $10$ fixed data splits provided by Pei et al. (2020), reporting the mean accuracy and standard deviation. Each split allocates $48\%$, $32\%$, and $20\%$ of the nodes per class to the training, validation, and test sets, respectively. All models are evaluated on the same set of splits. To ensure robust performance estimates and account for randomness in initialization and data splits, each hyperparameter configuration is evaluated using $10$ Monte Carlo repetitions. The set of hyperparameters is reported in Appendix B.1.

### 3.2.2 DATASETS

To evaluate model performance across diverse scenarios, we conduct experiments on homophilic graphs such as Cora McCallum et al. (2000), Citeseer Sen et al. (2008), and Pubmed Namata et al. (2012), and on heterophilic graphs such as Texas, Wisconsin, and Cornell from WebK, as well as Chameleon, Squirrel, and Film Rozemberczki et al. (2021); Tang et al. (2009). These datasets exhibit a wide range of homophily ratios, from **0.11** (highly heterophilic) to **0.81** (highly homophilic).

### 3.2.3 PERFORMANCE

Table 1 summarizes the performance of GRAND-ASC against other models. Our method demonstrates strong performance across both heterophilic and homophilic datasets, achieving state-of-the-art results on multiple benchmarks. On heterophilic datasets, GRAND-ASC achieves top performance on **Texas**, **Wisconsin**, **Squirrel**, and **Cornell**, while securing second place on **Chameleon** with a narrow margin of just $0.50\%$. On the **Film** dataset, GRAND-ASC remains highly competitive with 36.14%, placing within 0.75% of the top performer. Notably, GRAND-ASC also delivers strong results on homophilic datasets, achieving top-three performance on **Citeseer** (second best) and **Cora** (third best). The overall superiority of GRAND-ASC is reflected in its best-in-class mean rank of **2.6**, calculated by assigning each model a positional score on every dataset (1 for best, 2 for second best, etc.) and averaging across all nine datasets. This consistent performance across diverse graph types highlights the effectiveness of the adaptive skip connection mechanism in handling heterophilic datasets while maintaining acceptable performance on homophilic graph structures.

Table 1: Test accuracy and standard deviation over 10 experiments on each dataset. **Red** is the best, **Blue** the second best, and **Violet** the third best.

| | Texas | Wisconsin | Film | Squirrel | Chameleon | Cornell | Citeseer | PubMed | Cora | Mean Rank |
|---|---|---|---|---|---|---|---|---|---|---|
| Homophily | 0.11 | 0.21 | 0.22 | 0.22 | 0.23 | 0.30 | 0.74 | 0.80 | 0.81 | – |
| #Nodes | 183 | 251 | 7,600 | 5,201 | 2,277 | 183 | 3,327 | 18,717 | 2,708 | – |
| #Edges | 295 | 466 | 26,752 | 198,493 | 31,421 | 280 | 4,676 | 44,327 | 5,429 | – |
| #Classes | 5 | 5 | 5 | 5 | 5 | 5 | 7 | 3 | 6 | – |
| GCN | 60.54 ±5.57 | 54.90 ±5.95 | 28.13 ±1.14 | 27.26 ±1.33 | 37.94 ±2.43 | 45.14 ±5.28 | 75.47 ±1.55 | 87.31 ±0.55 | 86.52 ±1.14 | 10.5 |
| GAT | 61.89 ±5.85 | 55.10 ±3.45 | 28.69 ±0.99 | 32.02 ±2.10 | 45.02 ±2.13 | 47.84 ±7.84 | 74.83 ±1.08 | 86.15 ±0.49 | 85.77 ±1.09 | 9.7 |
| SAGE | 78.38 ±3.63 | 78.63 ±4.25 | 34.95 ±1.17 | 37.09 ±1.35 | 51.12 ±1.89 | 72.16 ±3.83 | 75.31 ±1.46 | 88.95 ±0.47 | 86.78 ±0.99 | 5.1 |
| Mixhop | 70.81 ±7.23 | 80.20 ±7.52 | 36.89 ±0.72 | 33.86 ±1.72 | 48.62 ±1.85 | 67.03 ±6.60 | 74.94 ±1.81 | 89.81 ±0.35 | 85.05 ±0.57 | 6.1 |
| GPS | 71.62 ±6.76 | 76.47 ±5.61 | 34.66 ±0.64 | 33.28 ±1.29 | 43.25 ±1.92 | 67.84 ±7.20 | 74.54 ±2.19 | 88.92 ±0.33 | 84.65 ±1.27 | 8.0 |
| DIRGNN | 84.59 ±7.93 | 80.00 ±4.94 | 36.66 ±1.08 | 48.06 ±2.87 | 62.83 ±2.00 | 70.27 ±4.36 | 73.97 ±1.81 | 89.83 ±0.35 | 84.39 ±1.00 | 5.2 |
| Jknet | 61.89 ±4.59 | 58.43 ±5.74 | 30.64 ±0.93 | 30.72 ±1.44 | 41.67 ±3.05 | 50.00 ±8.48 | 76.13 ±1.32 | 88.63 ±0.52 | 86.78 ±1.11 | 8.0 |
| GCNII | 67.57 ±10.1 | 80.78 ±5.39 | 36.77 ±0.65 | 35.13 ±1.72 | 48.93 ±1.72 | 64.05 ±8.11 | 76.09 ±1.68 | 89.80 ±0.43 | 87.67 ±0.98 | 4.6 |
| APPNP | 61.08 ±4.22 | 56.08 ±5.28 | 30.40 ±1.01 | 29.00 ±1.19 | 42.63 ±3.04 | 47.03 ±8.48 | 75.92 ±1.56 | 88.33 ±0.44 | 87.71 ±1.16 | 8.5 |
| GRAND++ | 81.62 ±6.14 | 80.78 ±4.45 | 34.05 ±1.23 | 52.83 ±2.99 | 71.27 ±2.21 | 71.35 ±5.16 | 75.16 ±1.97 | 86.71 ±0.63 | 85.69 ±1.23 | 5.4 |
| ACMP | 74.32 ±6.42 | 81.57 ±2.93 | 35.36 ±0.90 | 38.00 ±1.73 | 53.05 ±2.83 | 71.62 ±3.25 | 76.78 ±1.82 | 89.31 ±0.32 | 86.94 ±1.66 | 3.5 |
| GRAND-ASC | 86.76 ±3.91 | 85.49 ±5.13 | 36.14 ±1.23 | 62.09 ±5.96 | 70.77 ±1.65 | 73.51 ±3.78 | 76.18 ±1.24 | 87.92 ±0.50 | 87.12 ±1.00 | 2.6 |

We also compare our model to other continuous models, such as CGCN Xhonneux et al. (2020), GRAND Chamberlain et al. (2021a), BLEND Chamberlain et al. (2021b), and NSD Bodnar et al. (2022), with the results provided in Appendix B.2.

## 4 CONCLUSION

In this work, we introduced GRAND-ASC, a simple yet powerful linear continuous GNN that addresses the oversmoothing problem through an adaptive skip connection to the initial features. We provided a solid theoretical foundation, proving the model is well-posed, numerically stable, and formally bounds the Dirichlet energy away from zero. Our empirical results demonstrated that this approach is not only theoretically sound but also highly effective, achieving competitive state-of-the-art performance, particularly on heterophilic graphs. This indicates that complex nonlinear dynamics are not a prerequisite for preventing oversmoothing and that a carefully designed linear mechanism can yield powerful and robust graph learning. However, for simplicity, this paper considers the case where the attention matrix remains fixed during learning (i.e., time-independent). At the same time, the nonlinear version (where attention dynamically depends on node features) and the nonlinear version with rewiring (where the graph structure is adaptively pruned) can also be adapted to enhance the performance of GRAND-ASC. Another adaptive architecture worth studying is the case where different nodes are allowed to have different residual strengths, $\lambda_i(t)$; that is, the term $\lambda(t)$ in Equation (4) becomes $\mathbf{\Lambda}(t) = \text{diag}(\lambda_1(t), \dots, \lambda_n(t))$.

**Statement on Large Language Model Usage.** During the preparation of this work, the author(s) used ChatGPT to assist with proofreading and polishing the language of the manuscript. This was limited to correcting grammatical errors, improving sentence flow, and enhancing readability.

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

# A  APPENDIX: PROOFS

## A.1  THE PROOF OF THEOREM 1

*Proof.* For $k \in \arg\max_i x_i(t)$ we have

$$\frac{dx_k(t)}{dt} = \lambda(t) \sum_{j \in \mathcal{N}(k)} a(x_k, x_j)\big(x_j(t) - x_k(t)\big) + (1 - \lambda(t))\big(x_k(0) - x_k(t)\big).$$

Since $x_j(t) \le x_k(t)$ for every $j \in \mathcal{N}(k)$, the first (neighbor) term is non-positive, so

$$\frac{dx_k(t)}{dt} \le (1 - \lambda(t))\big(x_k(0) - x_k(t)\big).$$

Rearrange the inequality as

$$\frac{dx_k(t)}{dt} + (1 - \lambda(t))x_k(t) \le (1 - \lambda(t))x_k(0).$$

Multiplying by the integrating factor $\mu(t) := e^{\int_0^t (1 - \lambda(s))ds}$ gives

$$\frac{d}{dt}\big(\mu(t)x_k(t)\big) \le (1 - \lambda(t))\mu(t)x_k(0) = \frac{d\mu(t)}{dt}x_k(0).$$

Integrating from 0 to $t$ and using $\mu(0) = 1$ yields

$$\mu(t)x_k(t) - x_k(0) \leq x_k(0)\big(\mu(t) - 1\big).$$

Hence $\mu(t)x_k(t) \leq x_k(0)\mu(t)$, which implies $x_k(t) \leq x_k(0)$. A similar argument applies to the minimum case. Choose $l \in \arg\min_i x_i(t)$. Because $x_j(t) \geq x_l(t)$ for every $j \in \mathcal{N}(l)$, the neighbor term is nonnegative and

$$\frac{dx_l(t)}{dt} \geq (1 - \lambda(t))\big(x_l(0) - x_l(t)\big).$$

Multiplying by the same $\mu(t)$ and integrating yields $\mu(t)x_l(t) \geq x_l(0)\mu(t)$, which implies $x_l(t) \geq x_l(0)$. $\qquad\square$

## A.2 THE PROOF OF THEOREM 2

*Proof.* The system can be written in matrix form as

$$\mathbf{x}(t+1) = \mathbf{M}(t)\mathbf{x}(t) + \mathbf{c}(t)$$

where $\mathbf{M}(t) = (1 - \Delta t)\mathbf{I} + \Delta t\lambda(t)\mathbf{A}$ and $\mathbf{c}(t) = \Delta t(1 - \lambda(t))\mathbf{x}(0)$. Unrolling the recurrence gives

$$\mathbf{x}(t+1) = \left(\prod_{k=0}^{t} \mathbf{M}(t-k)\right)\mathbf{x}(0) + \sum_{j=0}^{t-1}\left(\prod_{k=0}^{j} \mathbf{M}(t-k)\right)\mathbf{c}(t-j-1) + \mathbf{c}(t).$$

Taking norms and applying submultiplicativity

$$\|\mathbf{x}^{(t+1)}\| \leq \left(\prod_{k=0}^{t} \|\mathbf{M}(t-k)\|\right)\|\mathbf{x}(0)\| + \sum_{j=0}^{t-1}\left(\prod_{k=0}^{j} \|\mathbf{M}(t-k)\|\right)\|\mathbf{c}(t-j-1)\| + \|\mathbf{c}(t)\|.$$

Using the fact that $\|\mathbf{A}\| = 1$ (since $\mathbf{A}$ is stochastic), and for any $t \geq 0$, $\|\mathbf{M}(t)\| \leq 1 - \Delta t + \Delta t\lambda(t) \leq 1 - \Delta t\delta = 1 - \xi$ where $\xi = \Delta t\delta > 0$. Also, for any $t \geq 0$, $\|\mathbf{c}(t)\| \leq \Delta t\|\mathbf{x}(0)\|$. Thus, we obtain

$$\|\mathbf{x}^{(t+1)}\| \leq (1 - \xi)^{t+1}\|\mathbf{x}^{(0)}\| + \Delta t\|\mathbf{x}^{(0)}\| \sum_{j=0}^{t-1}(1 - \xi)^{j+1} + \Delta t\|\mathbf{x}^{(0)}\|$$

$$= \left[(1 - \xi)^{t+1} + \Delta t\left((1 - \xi)\frac{1 - (1 - \xi)^t}{1 - (1 - \xi)} + 1\right)\right]\|\mathbf{x}^{(0)}\|.$$

Taking limsup (since we are not certain that the limit exists) from both sides, as $t \to \infty$, $(1-\xi)^t \to 0$, leaving

$$\limsup_{t\to\infty} \|\mathbf{x}^{(t)}\| \leq \Delta t\left(\frac{1 - \xi}{\xi} + 1\right)\|\mathbf{x}^{(0)}\| = \frac{\Delta t}{\xi}\|\mathbf{x}^{(0)}\|,$$

which concludes the proof. $\qquad\square$

## A.3 THE PROOF OF THEOREM 3

*Proof.* By Gelfand's spectral radius formula (see e.g., Horn & Johnson (2012))

$$\lim_{j\to\infty} \|\mathbf{M}^j\|^{1/j} = \rho(\mathbf{M}) < 1.$$

Choose $\gamma := \frac{\rho(\mathbf{M})+1}{2}$, so $\rho(\mathbf{M}) < \gamma < 1$. For $\epsilon := \gamma - \rho(\mathbf{M}) > 0$, there exists $N_0 \in \mathbb{N}$ such that for all $j \geq N_0$, $\|\mathbf{M}^j\|^{1/j} \leq \rho(\mathbf{M}) + \epsilon = \gamma$. Thus for all $j \geq N_0$,

$$\|\mathbf{M}^j\| \leq \gamma^j \quad . \tag{10}$$

For $j < N_0$, define

$$C' := \max_{0 \leq k < N_0} \|\mathbf{M}^k\|\gamma^{-k}.$$

Then, for all $j < N_0$,

$$\|\mathbf{M}^j\| = \left(\|\mathbf{M}^j\|\gamma^{-j}\right)\gamma^j \leq C'\gamma^j. \tag{11}$$

Combining Equation (10) and Equation (11) and letting $C := \max(1, C')$ proves the claim. $\qquad\square$

## A.4 THE PROOF OF THEOREM 4

*Proof.* The discrete system Equation (9) can be expressed in closed form as

$$\mathbf{x}(t+1) = R(\Delta t\mathbf{B})\mathbf{x}(t) + \boldsymbol{\Psi}(t), \tag{12}$$

where $R(z) = 1 + z + \frac{z^2}{2!} + \frac{z^3}{3!} + \frac{z^4}{4!}$ is the stability function of the RK4 method, and $\boldsymbol{\Psi}(t)$ is a constant term that accumulates the effect of the memory term over one time step, which is given by

$$\boldsymbol{\Psi}(t) = \frac{\Delta t}{6}\left(6\mathbf{I} + 3\Delta t\mathbf{B} + (\Delta t)^2\mathbf{B}^2 + \frac{(\Delta t)^3}{4}\mathbf{B}^3\right)\mathbf{c}(t).$$

Unrolling the recurrence relation Equation (12) over $t+1$ time steps yields

$$\mathbf{x}(t+1) = R^{t+1}(\Delta t\mathbf{B})\mathbf{x}(0) + \sum_{j=0}^{t} R^j(\Delta t\mathbf{B})\boldsymbol{\Psi}(t-j).$$

Taking the norm on both sides and applying the triangle inequality and submultiplicativity of the norm, we obtain

$$\|\mathbf{x}(t+1)\| \leq \|R(\Delta t\mathbf{B})\|^{t+1}\|\mathbf{x}(0)\| + \sum_{j=0}^{t}\|R(\Delta t\mathbf{B})\|^j\|\boldsymbol{\Psi}(t-j)\|. \tag{13}$$

To ensure asymptotic stability, we must show that $\|\mathbf{x}(t)\|$ remains bounded as $t \to \infty$. This requires that the spectral radius $\rho(R(\Delta t\mathbf{B})) < 1$, so that the first term vanishes exponentially.

Let $\mu$ be an eigenvalue of $\mathbf{B}$. Since $\mathbf{B} = \lambda(t)\mathbf{A} - \mathbf{I}$, we have $\mu = \lambda(t)\alpha - 1$, where $\alpha$ is an eigenvalue of $\mathbf{A}$. Given that $\|\mathbf{A}\|_2 \leq 1$, and $0 \leq \lambda(t) \leq 1$, it follows that $|\mu + 1| = |\lambda(t)\alpha| \leq \lambda(t)|\alpha| \leq \lambda(t) < 1$. Now, consider the scaled eigenvalue $\nu = \Delta t\mu$, with $\Delta t \in (0, 1]$. Then

$$|\nu + 1| = |\Delta t\mu + 1| = |\Delta t(\mu + 1) + (1 - \Delta t)| \leq \Delta t|\mu + 1| + (1 - \Delta t) < 1.$$

We now show that the RK4 stability function $R(z)$ satisfies $|R(\nu)| < 1$ for all $\nu$ such that $|\nu+1| < 1$. Note that the stability region of RK4 includes the disk $\{\nu \in \mathbb{C} : |\nu + 1| \leq 1\}$. This can be verified by checking key points. At $\nu = -1$: $R(-1) = \frac{3}{8} < 1$; at $\nu = -1 \pm i$: $R(-1 \pm i) = \frac{1}{6} \pm \frac{i}{3}$, so $|R(-1 \pm i)| = \frac{\sqrt{5}}{6} < 1$; at $\nu = -2$: $R(-2) = \frac{1}{3} < 1$. Since $R(z)$ is analytic and the boundary of the disk is mapped inside the unit circle, by the maximum modulus principle, $|R(\nu)| < 1$ for all $\nu$ with $|\nu+1| < 1$. Hence, $\rho(R(\Delta t\mathbf{B})) < 1$, and the first term in Equation (13) vanishes exponentially as $t \to \infty$.

We now bound the second term. First, note that

$$\|\boldsymbol{\Psi}(t)\| \leq \frac{\Delta t}{6}\left\|6\mathbf{I} + 3\Delta t\mathbf{B} + (\Delta t)^2\mathbf{B}^2 + \frac{(\Delta t)^3}{4}\mathbf{B}^3\right\|\|\mathbf{c}(t)\| \leq 3\|\mathbf{x}(0)\|,$$

where we used $\|\mathbf{B}\| \leq \|\lambda(t)\mathbf{A} - 1\| \leq 2$, $\Delta t \leq 1$, and $\|\mathbf{c}(t)\| = \|(1 - \lambda(t))\mathbf{x}(0)\| \leq \|\mathbf{x}(0)\|$. Also, as $\rho(R(\Delta t\mathbf{B})) < 1$, by Theorem 3, there exist constants $C > 0$ and $0 < \gamma < 1$ such that $\|R(\Delta t\mathbf{B})^j\| \leq C\gamma^j$ for all $j \geq 0$. Substituting into the second term of Equation (13)

$$\sum_{j=0}^{t}\|R(\Delta t\mathbf{B})\|^j\|\boldsymbol{\Psi}(t-j)\| \leq 3\|\mathbf{x}(0)\|\sum_{j=0}^{t}\|R(\Delta t\mathbf{B})\|^j$$

$$\leq 3\|\mathbf{x}(0)\|\sum_{j=0}^{t}C\gamma^j \leq 3C\|\mathbf{x}(0)\|\sum_{j=0}^{\infty}\gamma^j = \frac{3C\|\mathbf{x}(0)\|}{1 - \gamma}.$$

Thus, from Equation (13)

$$\|\mathbf{x}(t+1)\| \leq \|R(\Delta t\mathbf{B})\|^{t+1}\|\mathbf{x}(0)\| + \frac{3C\|\mathbf{x}(0)\|}{1 - \gamma}.$$

As $t \to \infty$, the first term vanishes and the proof is done. $\qquad\square$

## A.5 The Proof of Theorem 5

*Proof.* Since $\mathcal{L}$ is symmetric positive semi-definite, it admits an orthonormal eigenbasis $\{\mathbf{v}_1, \mathbf{v}_2, \ldots, \mathbf{v}_n\}$ where $\mathcal{L}\mathbf{v}_i = \gamma_i \mathbf{v}_i$. The first eigenvector is $\mathbf{v}_1 = \mathbf{D}^{1/2}\mathbf{1}$, corresponding to $\gamma_1 = 0$. Next, expand $\mathbf{x}(t)$ in this eigenbasis. The centering condition $\mathbf{x}(t)^\top \mathbf{D}^{1/2}\mathbf{1} = 0$ implies orthogonality to $\mathbf{v}_1$, so the expansion becomes

$$\mathbf{x}(t) = \sum_{i=2}^{n} c_i \mathbf{v}_i \quad \text{where} \quad c_i = \mathbf{x}(t)^\top \mathbf{v}_i.$$

Compute the quadratic form as

$$\mathcal{E}(\mathbf{x}(t)) = \sum_{i=2}^{n} \gamma_i c_i^2 \geq \gamma_2 \sum_{i=2}^{n} c_i^2 = \gamma_2 \|\mathbf{x}(t)\|_2^2,$$

where the equality follows from Parseval's identity and the inequality holds because $\gamma_i \geq \gamma_2$ for all $i \geq 2$. $\qquad\square$

## A.6 The Proof of Theorem 6

*Proof.* We begin by multiplying both sides of the dynamics Equation (3) by $x_i(t)$ and summing over all nodes $i$, which gives us

$$\sum_{i=1}^{n} x_i(t) \frac{\partial x_i(t)}{\partial t} = \lambda(t) \sum_{i=1}^{n} \sum_{j \in \mathcal{N}(i)} a(x_i, x_j)(x_j(t) - x_i(t)) x_i(t) + (1 - \lambda(t)) \sum_{i=1}^{n} (x_i(0) - x_i(t)) x_i(t).$$

The left-hand side simplifies to the time derivative of the squared $\ell_2$-norm

$$\sum_{i=1}^{n} x_i(t) \frac{\partial x_i(t)}{\partial t} = \frac{1}{2} \frac{\partial}{\partial t} \|\mathbf{x}(t)\|_2^2.$$

For the right-hand side, we analyze the two terms separately. The diffusion term can be rewritten as a sum over graph edges. For each undirected edge $(i, j)$, the contributions from both endpoints combine as follows

$$\lambda(t) \sum_{i=1}^{n} \sum_{j \in \mathcal{N}(i)} a(x_i, x_j)(x_j(t) - x_i(t)) x_i(t)$$

$$= \lambda(t) \sum_{(i,j) \in E} [a(x_i, x_j)(x_j(t) - x_i(t)) x_i(t) + a(x_j, x_i)(x_i(t) - x_j(t)) x_j(t)]$$

$$= \lambda(t) \sum_{(i,j) \in E} a(x_i, x_j) \left[ (x_j(t) - x_i(t)) x_i(t) + (x_i(t) - x_j(t)) x_j(t) \right]$$

$$= \lambda(t) \sum_{(i,j) \in E} a(x_i, x_j) \left[ x_j(t) x_i(t) - x_i(t)^2 + x_i(t) x_j(t) - x_j(t)^2 \right]$$

$$= -\lambda(t) \sum_{(i,j) \in E} a(x_i, x_j)(x_j(t) - x_i(t))^2.$$

The memory term simplifies directly

$$(1 - \lambda(t)) \sum_{i=1}^{n} (x_i(0) - x_i(t)) x_i(t) = (1 - \lambda(t)) \left( \langle \mathbf{x}(0), \mathbf{x}(t) \rangle - \|\mathbf{x}(t)\|_2^2 \right).$$

Combining these results, we obtain

$$\frac{1}{2} \frac{\partial}{\partial t} \|\mathbf{x}(t)\|_2^2 = -\lambda(t) \sum_{(i,j) \in E} a(x_i, x_j)(x_j(t) - x_i(t))^2 + (1 - \lambda(t)) \left( \langle \mathbf{x}(0), \mathbf{x}(t) \rangle - \|\mathbf{x}(t)\|_2^2 \right).$$

$$(14)$$

Consider the diffusion term in the last expression. The attention function $a(x_i, x_j)$ is bounded by some maximum value $a_{\max} \leq 1$ (a standard property of common attention mechanisms like softmax). Thus,

$$\sum_{(i,j) \in E} a(x_i, x_j)(x_j(t) - x_i(t))^2 \leq a_{\max} \sum_{(i,j) \in E} (x_j(t) - x_i(t))^2. \tag{15}$$

The resulting expression $\sum_{(i,j) \in E}(x_j(t) - x_i(t))^2$ is a well-known quadratic form that can be expressed using the graph Laplacian. Specifically, for any vector $\mathbf{x}(t)$, we have the identity

$$\sum_{(i,j) \in E} (x_j(t) - x_i(t))^2 = \mathbf{x}(t)^\top \mathbf{L} \mathbf{x}(t).$$

Combining this result with Equation (15), gives us

$$\sum_{(i,j) \in E} a(x_i, x_j)(x_j(t) - x_i(t))^2 \leq a_{\max} \mathbf{x}(t)^\top \mathbf{L} \mathbf{x}(t).$$

Since $\mathbf{L}$ is a symmetric positive semi-definite matrix, its quadratic form is bounded by its maximum eigenvalue, by the Rayleigh-Ritz theorem, as

$$\mathbf{x}(t)^\top \mathbf{L} \mathbf{x}(t) \leq \mu_{\max}(\mathbf{L}) \|\mathbf{x}(t)\|_2^2.$$

The Gershgorin Circle Theorem provides a practical bound on this maximum eigenvalue. For the Laplacian $\mathbf{L}$, the Gershgorin discs are centered at $\mathbf{L}_{ii} = d_i$ (the degree of node $i$) with radius $R_i = \sum_{j \neq i} |\mathbf{L}_{ij}| = d_i$. This implies that all eigenvalues $\mu$ of $\mathbf{L}$ satisfy $|\mu - d_i| \leq d_i$ for some $i$, and consequently $0 \leq \mu \leq 2d_i \leq 2d_{\max}$. Therefore

$$\mu_{\max}(\mathbf{L}) \leq 2d_{\max}.$$

Applying this eigenvalue bound and the fact that $a_{\max} \leq 1$ yields the final sequence of inequalities

$$\sum_{(i,j) \in E} a(x_i, x_j)(x_j(t) - x_i(t))^2 \leq a_{\max} \mathbf{x}(t)^\top \mathbf{L} \mathbf{x}(t) \leq a_{\max} \cdot 2d_{\max} \|\mathbf{x}(t)\|_2^2 \leq 2d_{\max} \|\mathbf{x}(t)\|_2^2.$$

Finally, as $\lambda(t) < 1$, we have

$$\lambda(t) \sum_{(i,j) \in E} a(x_i, x_j)(x_j(t) - x_i(t))^2 \leq 2d_{\max} \|\mathbf{x}(t)\|_2^2.$$

Substituting these bounds into Equation (14) yields the differential inequality

$$\frac{\partial}{\partial t} \|\mathbf{x}(t)\|_2^2 \geq -2 \left( 2d_{\max} + (1 - \lambda(t)) \right) \|\mathbf{x}(t)\|_2^2 + 2(1 - \lambda(t))\langle \mathbf{x}(0), \mathbf{x}(t) \rangle.$$

Let us define $\eta(t) := 2d_{\max} + (1 - \lambda(t))$ and $C(t) := 2(1 - \lambda(t))\langle \mathbf{x}(0), \mathbf{x}(t) \rangle$. This simplifies our inequality to

$$\frac{\partial}{\partial t} \|\mathbf{x}(t)\|_2^2 + 2\eta(t) \|\mathbf{x}(t)\|_2^2 \geq C(t).$$

Multiplying both sides of the last inequality by the integrating factor $\mu(t) = \exp\left( 2 \int_0^t \eta(s)ds \right)$ gives

$$\mu(t) \frac{\partial}{\partial t} \|\mathbf{x}(t)\|_2^2 + 2\eta(t)\mu(t) \|\mathbf{x}(t)\|_2^2 \geq \mu(t)C(t).$$

As the left-hand side is the derivative of $\mu(t)\|\mathbf{x}(t)\|_2^2$, we have

$$\frac{\partial}{\partial t} \left[ \mu(t)\|\mathbf{x}(t)\|_2^2 \right] \geq \mu(t)C(t).$$

Integrating both sides from $0$ to $t$ yields

$$\mu(t)\|\mathbf{x}(t)\|_2^2 - \mu(0)\|\mathbf{x}(0)\|_2^2 \geq \int_0^t \mu(s)C(s)ds.$$

Since $\mu(0) = 1$, we have

$$\mu(t)\|\mathbf{x}(t)\|_2^2 \geq \|\mathbf{x}(0)\|_2^2 + \int_0^t \mu(s)C(s)ds.$$

Dividing through by $\mu(t)$

$$\|\mathbf{x}(t)\|_2^2 \geq \exp\left(-2\int_0^t \eta(s)ds\right)\|\mathbf{x}(0)\|_2^2 + \int_0^t C(s)\exp\left(-2\int_s^t \eta(r)dr\right)ds.$$

We now apply the theorem's assumptions: $1 - \lambda(s) \geq \delta > 0$ and $\langle \mathbf{x}(0), \mathbf{x}(s) \rangle > m > 0$. This gives us $C(s) \geq 2\delta m$. Also, since $1 - \lambda(t) \leq 1$, we have $\eta(r) \leq 2d_{\max} + 1$. These bounds give us

$$\exp\left(-2\int_0^t \eta(s)ds\right) \geq \exp\left(-2(2d_{\max}+1)t\right), \quad \exp\left(-2\int_s^t \eta(r)dr\right) \geq \exp\left(-2(2d_{\max}+1)(t-s)\right).$$

Substituting these bounds yields

$$\|\mathbf{x}(t)\|_2^2 \geq \|\mathbf{x}(0)\|_2^2 e^{-2(2d_{\max}+1)t} + \int_0^t 2\delta m e^{-2(2d_{\max}+1)(t-s)}ds.$$

Evaluating the integral as

$$\int_0^t 2\delta m e^{-2(2d_{\max}+1)(t-s)}ds = 2\delta m e^{-2(2d_{\max}+1)t}\int_0^t e^{2(2d_{\max}+1)s}ds = \frac{\delta m}{2d_{\max}+1}\left(1 - e^{-2(2d_{\max}+1)t}\right).$$

Thus, we obtain the following explicit lower bound

$$\|\mathbf{x}(t)\|_2^2 \geq \|\mathbf{x}(0)\|_2^2 e^{-2(2d_{\max}+1)t} + \frac{\delta m}{2d_{\max}+1}\left(1 - e^{-2(2d_{\max}+1)t}\right).$$

Taking the limit as $t \to \infty$, the exponential terms vanish, yielding the asymptotic lower bound

$$\lim_{t\to\infty}\|\mathbf{x}(t)\|_2^2 \geq \frac{\delta m}{2d_{\max}+1}.$$

Since $\mathbf{x}(t)$ is mean-centered by assumption, we apply Theorem 5, which states $\mathcal{E}(\mathbf{x}(t)) \geq \gamma_2\|\mathbf{x}(t)\|^2$. Combining these results completes the proof

$$\mathcal{E}(\mathbf{x}(t)) \geq \gamma_2 \cdot \frac{\delta m}{2d_{\max}+1} = \frac{\delta m \gamma_2}{2d_{\max}+1}.$$

$\square$

### A.7 THE PROOF OF COROLLARY 1

In this section, we analyze the asymptotic behavior of GRAND-ASC under the assumption that $\lambda(t) = \lambda$ is constant. We demonstrate that, unlike the standard GRAND, the solution does not converge to a constant vector. Instead, it depends on both the graph structure (via the Laplacian) and the parameter $\lambda$. We begin by rewriting the GRAND-ASC dynamics as

$$\frac{d\mathbf{x}(t)}{dt} = -\mathbf{M}\mathbf{x}(t) + \mathbf{b}, \tag{16}$$

where $\mathbf{M} = \lambda\mathcal{L} + (1-\lambda)\mathbf{I}$ and $\mathbf{b} = (1-\lambda)\mathbf{x}(0)$. Note that for $\lambda \in (0,1)$, and since $\mathcal{L}$ is the normalized Laplacian matrix, the eigenvalues of $\mathbf{M}$ are strictly positive. This follows because $\mathbf{M}$ shifts the eigenvalues of $\mathcal{L}$ by $1 - \lambda > 0$, ensuring that $\mathbf{M}$ is strictly positive definite and hence invertible.

The solution to Equation (16) is given by

$$\mathbf{x}(t) = e^{-\mathbf{M}t}\mathbf{x}(0) + \int_0^t e^{-\mathbf{M}(t-s)}\mathbf{b}ds$$

Evaluating the integral yields

$$\int_0^t e^{-\mathbf{M}(t-s)}\mathbf{b}ds = \mathbf{M}^{-1}(\mathbf{I} - e^{-\mathbf{M}t})\mathbf{b}.$$

Thus, the solution simplifies to

$$\mathbf{x}(t) = e^{-\mathbf{M}t}\mathbf{x}(0) + \mathbf{M}^{-1}(\mathbf{I} - e^{-\mathbf{M}t})\mathbf{b}$$

As $t \to \infty$, $e^{-\mathbf{M}t} \to \mathbf{0}$ (since $\mathbf{M}$ is strictly positive definite), and therefore $\mathbf{x}(t) \to (1-\lambda)\mathbf{M}^{-1}\mathbf{x}(0)$.

# B  APPENDIX: EXPERIMENTS

## B.1  HYPERPARAMETER SETUP

We evaluate the performance of our model across a comprehensive range of hyperparameters within a predefined search space. Specifically, we consider learning rates of $\{10^{-1}, 10^{-2}\}$, weight decays of $\{10^{-3}, 10^{-4}, 10^{-5}, 10^{-6}, 10^{-7}\}$, hidden dimensions of $\{8, 16, 32, 64\}$, and numbers of hidden layers from $\{3, 5, 7\}$. The dropout rate is varied among $\{0.0, 0.1, 0.2, 0.4\}$. For our model, we use a fixed number of $4$ attention heads, and the integration time $T$ is chosen from $\{1, 2\}$. All models are trained for a maximum of 300 epochs with an early-stopping patience of 30 epochs. For heterophilic datasets, we employ a one-layer MLP as the encoder and decoder in GRAND-ASC, whereas for homophilic datasets, we use a single-layer GCN.

## B.2  MORE EXPERIMENTS

Table 2 compares GRAND-ASC with leading continuous models, where baseline results are taken from Bodnar et al. (2022). Despite employing a more constrained hyperparameter search, GRAND-ASC demonstrates highly competitive performance and achieves state-of-the-art results on four of the six heterophilic datasets: Texas, Wisconsin, Squirrel, and Chameleon. On the Film dataset, it delivers the second-best performance, trailing the top model by only $1.14\%$. On homophilic datasets (Cora, Citeseer, PubMed), where performance is already high across many methods, GRAND-ASC remains competitive, with differences from the best model being marginal (less than $1\%$ in absolute accuracy, as shown in Figure 2). *These results demonstrate that GRAND-ASC is a robust general-purpose model that excels particularly in heterophilic settings while maintaining acceptable performance on homophilic graphs.*

Table 2: Test accuracy and standard deviation over 10 experiments on each dataset with **continuous** models, using different train/validation/test splits. **Red** is the best, **Blue** the second best.

|  | Texas | Wisconsin | Film | Squirrel | Chameleon | Cornell |
|---|---|---|---|---|---|---|
| NSD | **83.78**±**6.62** | **85.29**±**3.31** | **37.28**±**0.74** | **52.57**±**2.76** | **66.40**±**2.28** | **84.60**±**4.69** |
| BLEND | 83.24±4.65 | 84.12±3.56 | 35.63±0.89 | 43.06±1.39 | 60.11±2.09 | **85.95**±**6.82** |
| GRAND | 75.68±7.25 | 79.41±3.64 | 35.62±1.01 | 40.05±1.50 | 54.67±2.54 | 82.16±7.09 |
| CGNN | 71.35±4.05 | 74.31±7.26 | 35.95±0.86 | 29.24±1.09 | 46.89±1.66 | 66.22±7.69 |
| GRAND-ASC | **86.76** ±**3.91** | **85.49** ±**5.13** | **36.14** ±**1.23** | **62.09** ±**5.96** | **70.77** ±**1.65** | 73.51 ±3.78 |

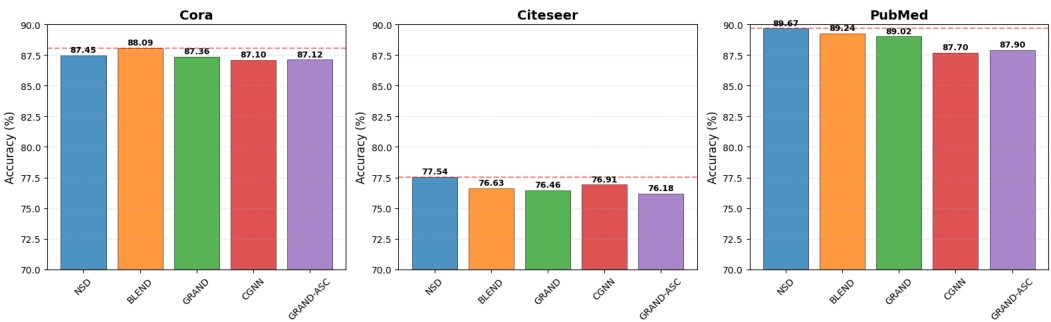

Figure 2: Accuracy of the continuous GNNs on hemophilic datasets.