# OpenReview forum: "Graph Neural Diffusion with Adaptive Skip Connection"
_ICLR.cc/2026/Conference — ICLR 2026 Conference Withdrawn Submission_

### Official Review · Reviewer_Dzhr · 2025-10-25

**Soundness:** 3
**Presentation:** 3
**Contribution:** 3
**Rating:** 4
**Confidence:** 3

**Summary:**

The paper proposes GRAND-ASC, a continuous-depth GNN obtained by adding an adaptive skip connection from the initial features $X(0)$ to GRAND-style diffusion. The dynamics are
$$
\frac{\partial X(t)}{\partial t}=\lambda(t)\mathrm{div}\big[G(X(t),t)\nabla X(t)\big]+(1-\lambda(t))\big(X(0)-X(t)\big),
$$
where a learnable, time-dependent scalar $\lambda(t)\in(0,1)$ trades off neighborhood smoothing and feature preservation. The paper proves well-posedness, numerical stability for Euler and RK4 discretizations, and a lower bound on Dirichlet energy that prevents oversmoothing. Empirically, GRAND-ASC attains strong results on heterophilic benchmarks (Texas, Wisconsin, Squirrel, Cornell, etc.) while remaining competitive on homophilic graphs; a large comparison table shows favorable mean rank across nine datasets.

**Strengths:**

* Simple, principled dynamics that directly counter diffusion’s collapse; clean theory (well-posedness, Euler/RK4 stability, energy floor).
* Strong heterophily performance and competitive homophily results; comprehensive baseline table on standard splits.
* Parameter efficiency (learn a scalar $\lambda(t)$ per layer) with an interpretable smoothing–memory trade-off.

**Weaknesses:**

1. Equations use $a(x_i(t),x_j(t))$ (time-varying), but text elsewhere says attention is fixed (time-independent) for linearity and theory. Which is used in experiments and theorems? Please reconcile and report both variants.
2. The oversmoothing guarantee needs $\langle x(0),x(t)\rangle>0$ and $1-\lambda(t)\ge\delta$; show how often these hold in training, and what happens when $\lambda(t)\to 1$. Provide monitoring plots.
3. Missing ablations for constant vs. adaptive $\lambda(t)$; Euler vs. RK4; step size $\Delta t$; and runtime/memory vs. APPNP/GCNII/GRAND++.
4. Energy analysis scope. The energy lower bound is stated for centered features and connected graphs; discuss behavior under class imbalance, weak connectivity ($\gamma_2$ small), or high degree skew $(d_{\max})$.

**Questions:**

1. Which attention is used? Are the reported results with time-independent attention (frozen from $X(0)$) or time-dependent $A(X(t))$? If the latter, is the system still linear in practice? Please provide a side-by-side ablation.
2. $\lambda(t)$ behavior. How does the learned $\lambda(t)$ evolve over depth and datasets (heterophilic vs. homophilic)? Report per-layer trajectories and correlate with accuracy and Dirichlet energy.
3. Beyond theorems, do you observe training instabilities as $\lambda(t)$ approaches 1 or 0? Any regularizers to keep $1-\lambda(t)\ge\delta$?
4. Can the energy lower bound be extended to allow occasional negative $\langle x(0),x(t)\rangle$ while still preventing collapse?

---

### Official Review · Reviewer_t7Qg · 2025-10-31

**Soundness:** 2
**Presentation:** 3
**Contribution:** 2
**Rating:** 2
**Confidence:** 5

**Summary:**

This paper proposes a very simple and well-known method (GRAND+Adaptive Skip Connection), an ODE-based graph neural network that modulates the diffusion rate via a learnable skip coefficient $\lambda(t)$. While the formulation is mathematically clean and the theoretical analysis is sound, the methodological novelty is limited and the empirical evaluation is insufficient to justify the claimed contributions.

**Strengths:**

1. Theoretical analysis is well written, covering existence, uniqueness, and stability of the diffusion dynamics. Proofs are technically correct.
2. Figures and equations are neatly organized, and the derivation is easy to follow.
3. The adaptive skip term adds negligible computational overhead.

**Weaknesses:**

1. The adaptive skip connection is a minor variation of existing residual or teleport mechanisms in diffusion-based GNNs (e.g., APPNP, GCNII, GRAND++). This does not fundamentally alter the diffusion dynamics; it simply introduces a tunable linear interpolation between identity and diffusion terms.
2. All evaluations are on small citation or heterophilic datasets (Cora, Chameleon, Squirrel, etc.), which are no longer sufficient for demonstrating robustness or scalability. There is no validation on large-scale and realistic datasets such as OGB-Arxiv, OGB-Products, or OGB-Proteins, where the benefits of stable continuous diffusion could be more evident. Without these results, it remains unclear whether GRAND-ASC actually scales or generalizes.
3. There is no analysis of how $\lambda(t)$ evolves during training or why adaptivity helps. An ablation comparing fixed vs. learnable $\lambda$ would be necessary to isolate its contribution.
4. Reported improvements over GRAND++ and ACMP are within 1–2%, which is within typical variance and not statistically convincing.
5. The claim of “competitive SOTA” seems inflated given the simplicity of the modification.
6. Bounding the Dirichlet energy away from zero is not new; similar results exist for residual or constrained diffusion systems.
7. Proofs are correct but largely algebraic restatements of known lemmas (e.g., spectral gap lower bounds).
8. The introduction frames nonlinearity as the only known way to fight oversmoothing — that’s inaccurate. Prior works (GCNII, APPNP, DAGNN, JKNet, DeepGCN) already demonstrated skip-based linear fixes. The only “continuous” element here is embedding this skip in a diffusion ODE.
9. The comparison to GRAND++ (Thorpe et al. (2022)) doesn’t convincingly justify the gap. And I'm surprised why Table 1 doesn't even compare GRAND (I saw it in the appendix). And the following models mentioned in Line 168-169: GRAND-based models Chamberlainetal.(2021a); Thorpeetal.(2022); Wang et al.(2023); **Li et al.(2024b).** were not fully compared.
10. The paper claims that GRAND-ASC mitigates over-smoothing, yet no experiments varying depth or diffusion horizon are provided to substantiate this.

Minor:

You can use \citep{} to cite your reference.

**Questions:**

1. The experiments are all on small citation graphs (Cora, Chameleon, Squirrel). How does GRAND-ASC scale to large datasets such as OGB-Arxiv or OGB-Products?
2. The continuous-time formulation could suffer from numerical instability or over-smoothing for large $\lambda(t)$. How sensitive are results to integration step size or ODE solver tolerance?
3. Were the hyperparameters for baselines (especially GRAND++ and GCNII) re-tuned on the same splits? Some reported gains (≈1–2 %) may fall within variance bounds.

---

### Official Review · Reviewer_SWEX · 2025-10-31

**Soundness:** 2
**Presentation:** 3
**Contribution:** 3
**Rating:** 4
**Confidence:** 3

**Summary:**

This paper proposes an architecture for addressing the common problem of oversmoothing in diffusion-based architectures. In such architectures, local averaging over neighborhoods leads to smoothing, an effect that becomes stronger when applied sequentially, as in deep architectures. In essence, the proposed approach supplements local averaging by retaining a proportion of the original signal through a linear (convex) combination of the two, thereby preventing degeneration to a constant. The implementation of this approach, called *Adaptive Skip Connection*, builds on Graph Neural Diffusion, where an adaptive (learnable) proportion of the input signal is preserved.  The authors provide both a theoretical analysis of the approach and numerical evaluations on node classification tasks, demonstrating the efficacy of the proposed method.

**Strengths:**

I like many aspects of this paper. It is overall well written and pleasant to read. It is well organized, and I appreciate that the related work section appears after the detailed introduction, at a point where both the problem and the approach have been described, making it more natural to discuss connections to existing work.

The authors present a simple approach to an important problem, and I appreciate that they are upfront about the method being simple yet effective.

**Weaknesses:**

**(1) Theoretical Foundation (Section 2):**
I can’t help but feel that the authors, concerned about the simplicity of the proposed approach, sought to provide additional analysis to justify it. While the theoretical analysis appears solid (aside from some technical and notational issues discussed below), it seems to reflect a common approach to signal filtering, where the output of a low-pass filter is linearly combined with its input to control its intensity, and is closely related to a straightforward analysis of a damped variant of the heat equation.

The analysis that follows, including Theorems 1 and 2, and to some extent Theorems 5, 6, and Corollary 1, strikes me as rather straightforward. I would not be surprised if existing results (in related papers or textbooks) already provide similar analysis for Eqs. (3) and (4). Theorem 3 also seems to follow directly from the submultiplicativity of matrix norms. It is possible that I am missing something fundamental here.

Overall, Section 2 (currently three pages) feels disproportionately substantial given the level of contribution and novelty it provides.

---

**(2) Technical Inaccuracies and Inconsistencies:**
- *Line 100:* $\mathrm{x}_i$ denotes the feature vector of the $i$-th node (a row of $\mathrm{X}$). Later, in line 135, the same notation $\mathrm{x}_i$ is used for the $i$-th feature (a column of $\mathrm{X}$). Subsequently, the index is omitted, and $\mathrm{x}$ denotes a general feature vector of dimension $n$.
- $x_i$ is the $i$-th element of $\mathrm{x}$ and therefore a scalar. However, in the definition of $a(x_i, x_j)$ in line 149, it appears to be a vector of dimension $d_k$. $d_k$ is the “feature dimension per head”, which I found to be unclear as well.
- *Equation (3):* It is confusing that $x_i$ appears both with time dependence $x_i(t)$ and without it. Similarly, are $\mathbf{W}_Q$ and $\mathbf{W}_K$ time-dependent?
- *Theorem 1:* Should these be $\arg \min$ and $\arg \max$ instead of $\min$ and $\max$? Namely, is the value of $x_i(t)$ bounded between the *indices* of the minimal and maximal entries of $\mathrm{x}(0)$?
- *Equation (5):* Should it read $x_i(t + \Delta t)$?

---

**(3) Experiments:**
- The synthetic experiment (Section 3.1) focuses on the Dirichlet energy to demonstrate that the proposed approach avoids oversmoothing. However, it does not show that any meaningful evolution (message passing) occurs. For example, the identity network would score highly in this evaluation despite contributing nothing to feature evolution.
- What exactly is the node classification problem (how is it defined, what data is used, and what is its objective)? What is the evaluation metric? What do the experiments show? The results in Table 1 are difficult to interpret without this context.

**Questions:**

Overall, I have mixed feelings about this work. I like several aspects of it, but in light of the issues above, I lean against acceptance. I am open to revising my score if the authors provide compelling answers to my concerns above, or if other reviews reveal that I have fundamentally misunderstood the paper.

**Additional comments and questions:**
* **Line 62:** “improving the performance of the continuous models is generally difficult as they must use a single set of weights throughout the entire integration time” — unclear; please explain or revise.
* **Line 105:** “eigenvalue spectrum is constrained” — do you mean *bounded*?
* **Line 235:** “obtaining an analytic solution for GRAND-ASC is challenging due to its time-dependent dynamics” — is it *challenging* or *impossible/infeasible*?
* **Lines 266–269:** Why not introduce this simplification earlier, perhaps directly into Equation (3)?

---

### Official Review · Reviewer_MgHc · 2025-10-31

**Soundness:** 2
**Presentation:** 3
**Contribution:** 1
**Rating:** 2
**Confidence:** 3

**Summary:**

The paper proposes GRAND-ASC, a modification to the continuous-depth GRAND model that addresses the oversmoothing problem in GNNs. The key idea is to add a skip connection to the initial node features combined with a learnable, time-dependent scalar $\lambda(t)$ that adaptively balances diffusion and feature preservation. The authors claim theoretical guarantees of well-posedness, numerical stability, and a lower bound on the Dirichlet energy, suggesting the model mitigates oversmoothing without introducing nonlinearities. Experiments on standard node classification benchmarks (homophilic and heterophilic) are used to demonstrate empirical competitiveness.

**Strengths:**

1. The paper is clear and easy to follow.

2. The introduction of an adaptive skip connection is conceptually simple. This makes the approach easily reproducible and potentially applicable to other continuous-depth GNN frameworks.

3. The paper provides mathematical justification for stability and boundedness.

**Weaknesses:**

1. The core contribution, i.e., adding an adaptive skip connection to GRAND, is extremely incremental. Skip connections to initial features are well-established (PPNP, GCNII, DAGNN, DRGCN, etc.), and time-dependent or layer-adaptive weights have been explored in multiple works (e.g., Zhang et al., 2023; Liu et al., 2021). The “adaptive $\lambda(t)$” adds almost no conceptual novelty beyond these prior formulations. The paper reads like a minor variation dressed with unnecessary mathematical complexity.

2. The theoretical guarantees section is overstated. The stability proofs rely on standard ODE theory and offer no new insights beyond textbook results. The bounded Dirichlet energy analysis is mathematically correct but tautological. If you constrain $\lambda(t) \in (0,1)$ and use a linear diffusion operator, you necessarily prevent collapse to a constant solution. The claimed “theoretical foundation” is thus trivial rather than groundbreaking.

3. The claimed state-of-the-art results are marginal and statistically questionable. Improvements are within the standard deviation on many datasets, undermining claims of clear superiority.

4. The heterophilic datasets used are small and over-recycled, there’s no evaluation on larger or more realistic graphs (e.g., OGB datasets).

5. The adaptive $\lambda(t)$ is described as time-dependent, but experiments do not convincingly show how its learned dynamics evolve, i.e., no ablation, visualization, or analysis of $\lambda(t)$ trajectories is provided.

**Questions:**

1. How is GRAND-ASC fundamentally different from GCNII (2020) or PPNP (2019) and other variants, which also use initial skip connections and adaptive weighting? Beyond the continuous-time formulation, what unique property does GRAND-ASC introduce?

2. Can the authors show how $\lambda(t)$ evolves during training (e.g., learned curves, layer-wise profiles)?

3. Continuous solvers (even with RK4) are expensive. How does GRAND-ASC scale compared to discrete GNNs or even GRAND++ in terms of training time, memory?

---

### Note · Authors · 2025-11-13

I have read and agree with the venue's withdrawal policy on behalf of myself and my co-authors.